# Cost-Effective Detection of Perfluoroalkyl Carboxylic Acids with Gas Chromatography: Optimization of Derivatization Approaches and Method Validation

**DOI:** 10.3390/ijerph17010100

**Published:** 2019-12-21

**Authors:** Zhen Li, Hongwei Sun

**Affiliations:** 1Key Laboratory of Pesticide & Chemical Biology of Ministry of Education, Institute of Environmental & Applied Chemistry, College of Chemistry, Central China Normal University, Wuhan 430079, China; lizz0920@163.com; 2Hubei High-Tech Innovation and Business Incubation Center, Wuhan 430000, China; 3Division of Environmental Science and Engineering, Pohang University of Science and Technology (POSTECH), Pohang 37673, Korea

**Keywords:** perfluoroalkyl substance, PFOA, gas chromatography, derivatization, solid phase extraction

## Abstract

The reliable quantification of perfluoroalkyl carboxylic acids (PFCAs) in environmental samples like surface water by using gas chromatography (GC) remains challenging because the polar PFCAs call for derivatization before injection and problems involving the integration of sample pretreatment and derivatization procedures. Here we proposed a cost-effective method for the GC based determination of C4–C12 PFCAs in surface water samples by integrating solid phase extraction and PFCAs anilide derivatization. First, we assessed the performance of different PFCAs derivatization methods, namely esterification and amidation. Esterification was unable to derivatize C4–C6 PFCAs. On the contrary, amidation procedures by using 2,4-difluoroaniline (DFA) and *N*,*N*′-dicyclohexylcarbodiimide (DCC) could successfully transform all the PFCA analogs to produce anilide derivatives, which could be easily detected by GC. Then the reaction conditions in the amidation approach were further optimized by using orthogonal design experiments. After optimizing the instrumental parameters of GC, the limits of detection (LOD) of this derivatization method were determined to be 1.14–6.32 μg L^−1^. Finally, in order to establish an intact method for the quantification of PFCAs in surface water samples, solid phase extraction (SPE) was used for extraction and cleanup, which was further integrated with the subsequent amidation process. The SPE-amidation-GC method was validated for application, with good accuracy and precision reflected by the PFCAs recoveries and derivatization of triplicates. The method reported here could provide a promising and cost-effective alternative for the simultaneous determination of C4–C12 PFCAs in environmental water samples.

## 1. Introduction

Perfluoroalkyl acids (PFAAs) refer to a series of aliphatic acids with all the H atoms in the C–H bonds replaced by F atoms. According to different functional groups, they are mainly divided into two categories: perfluoroalkyl carboxylic acids (PFCAs) and perfluoroalkyl sulfonates (PFSAs) [1,2]. PFAAs and their precursors are widely used as a surfactant in metal plating, ingredients of polytetrafluoroethylene and polyvinylidene fluoride plastics, surface coatings for textile and paper, components of fire-fighting foam, cosmetics, textiles, carpets, and leather [3,4,5]. In recent years, it has been reported that PFAAs are environmentally-persistent, long-distance migratory, bio-accumulative, and highly toxic. PFAAs are recalcitrant to natural degradation due to the highly polarized C–F bonds and steric hindrance for the attack of external reactants [6,7]. Long-distance migration is manifested by their ubiquitous occurrence around the world, including polar regions [8]. PFAAs containing more than seven carbon atoms showed bioaccumulation factor (BCF) exceeding 1000 L·kg^−1^ [9]. In addition, as a peroxisome proliferator in organisms, PFCAs can cause hepatotoxicity [10,11]. PFCAs were also reported to adversely affect the development, immunity, and endocrine of organisms [12]. What is worse, unlike the legacy persistent organic pollutants (POPs) such as dichloro-diphenyl-trichloroethanes (DDTs) and hexachlorocyclohexane (HCHs), which are highly lipophilic, PFAAs are basically water soluble, and this might increase their bioavailability and the risk of exposure to aquatic species.

Regarding the detection methods for PFCAs, the most common methods used currently are high-performance liquid chromatography-tandem mass spectrometry (HPLC-MS/MS). Despite the advantages of accuracy and the simple procedure for sample preparation, the application of the HPLC-MS/MS is still limited due to the high cost of the instruments, especially in less developed areas [13]. Moreover, the background problems caused by the leakage of PFAAs from the fluorinated polymers in HPLC-MS systems also pose challenges to this method [14,15]. Therefore, developing both reliable and cost-effective methods of PFCAs analysis is still vital for the extensive research of PFCAs. Compared with HPLC-MS/MS, gas chromatography (GC) is relatively cheaper and easier for access, which has been widely adopted for the analysis of legacy POPs, such as organochlorine pesticides [16], polyaromatic hydrocarbons [17], polychlorinated biphenyls, and polybrominated diphenyl ethers [18]. In addition, GC equipped with a micro-electron capture detector (μECD) is especially useful for the detection of halogen-containing substances like PFCAs with ultrahigh sensitivity. However, reports about the application of GC for the detection of PFCAs are rare because PFCAs are in the form of anion at neutral pH, which makes these compounds difficult to evaporate for GC analysis because of their low volatility [19,20]. Therefore, the PFCA anions need to be derivatized before injection into GC in order to lower their boiling points and improve the thermal stability [21,22].

As early as the 1980s, Belisle and Hagen described a GC method for PFCAs analysis. The PFCAs were converted to their methyl esters by reaction with diazomethane [23]. Although this derivatization process could generate relatively good results, the use of diazomethane, which is both exceedingly toxic and explosive, made the method unsuitable for extensive application. Langlois et al. reported the use of acidic *iso*-propanol as the esterification agent instead of diazomethane [24]. This method showed qualitative derivatization of perfluorooctane sulfonate (PFOS) and PFCAs with more than six carbon atoms. However, the derivatization performance of PFCAs with shorter carbon chains was not mentioned. However, with the restrictions on the production and use of eight-carbon perfluorocarbon products by international organizations, short-chain PFAAs are gradually being widely used as substitutes, and therefore, the need to determine these PFAAs in environmental samples are increasing [25]. Moreover, Langlois et al. merely reported the derivatization of PFCAs standards, and the combination of such method with the pretreatment procedures of environmental samples such as surface water was not studied. In field analysis applications, the pretreatment procedures are of great importance because PFCAs are generally of trace concentrations in environmental samples; thus, extraction and pre-concentration of these compounds are necessary. In addition, the complex matrices in the samples might pose severe interference for the accurate analysis of PFCAs; therefore, cleanup procedures have to be introduced to reduce the matrix effect. Scott et al. reported the production of 2,4-difluoroanilides of PFCAs for subsequent GC-MS detection [26]. This method could achieve relatively good derivatization efficiencies for PFCA standards (C3–C9), and also, the authors managed to analyze the PFCAs in surface water samples by using this derivatization strategy. Basically, 1 L of surface water sample was directly reduced to 50 mL on a rotary evaporator, followed by the derivatization procedure in mixed solvent of water and ethyl acetate. Obviously, this method is time consuming because of the evaporation of a large volume of water. Meanwhile, the interference of the water matrix cannot be ruled out since no cleanup procedures were included. Alternatively, they introduced the XAD-resin extraction and cleanup for better quantification, but the recoveries of the target PFCAs were not satisfying. Actually, only the recoveries of three target PFCAs (C7–C9) met the standard for quantitative analysis, and the recoveries of other PFCAs congeners were either extremely low (around 30%) or non-available. Overall, it can be concluded that it remains challenging to quantify both the short- and long-chained PFCAs in environmental samples using derivatization and the GC method, especially when the derivatization has to be integrated with the sample pretreatment procedures.

Therefore, in the present study, we aim to develop a complete GC-based method for the easy detection of both short- and long-chained PFCAs in water samples. This method should integrate both the derivatization and sample pretreatment procedures, including the extraction, cleanup, and pre-concentration of target PFCAs from surface water samples, as well as the subsequent derivatization procedure and GC instrumental analysis method. This study will provide a useful and cost-effective alternative of the HPLC-MS-MS approach for the quantitative analysis of PFCAs in water samples.

## 2. Materials and Methods

### 2.1. Chemicals and Materials

Nine PFCAs standards used in this study were purchased from Sigma-Aldrich (St. Louis, Missouri, USA). The details, including compound name, formula, acronym, and CAS No., are provided in Table 1. The stock solution of single standard was prepared at a concentration of 1000 mg L^−1^ in methanol, and they were further mixed and diluted with methanol to obtain the mixed standards, with each compound at the concentration of 100 mg L^−1^. The standard samples were stored at 4 °C before use. Dichloromethane (DCM), n-hexane, ethyl acetate (EAC), ether, acetonitrile, methanol (MeOH), ethanol, *iso*-propanol (IPA), tetrabutyl-ammonium hydrogen sulfate (TBASH) are HPLC grade, and they were purchased from Thermo Fisher (Waltham, Massachusetts, USA). Hydrochloric acid and sulfuric acid are analytical grade, purchased from Sinopharm Chemical Reagent Co., Ltd, China. Analytical grade sodium chloride (NaCl), sodium hydroxide (NaOH), sodium bicarbonate (NaHCO_3_), anhydrous sodium sulfate (Na_2_SO_4_), ammonium acetate (NH_4_Ac), ammonium hydroxide (NH_4_OH), and hydrochloric acid (HCl) were purchased from Shanghai Chemical Co., Ltd, China. *N*,*N*-dicyclohexylcarbodiimide (DCC) and 2,4-difluoroaniline (DFA) were purchased from Braunway Technology Co., Ltd (Beijing, China). CNWBOND WAX SPE Cartridges (3 mL/60 mg) were purchased from ANPEL Laboratory Technologies (Shanghai, China) Inc.

### 2.2. Instruments

An Agilent 7820A gas chromatography equipped with a μECD detector (Agilent Technologies, Santa Clara, CA, USA) was used for the qualification and quantification of PFCAs derivatives. The GC column was HP-5 with dimensions of 30 m × 0.32 mm × 0.25 μm. Nitrogen was adopted as the carrier gas at a flow rate of 1 mL·min^−1^. The inlet was operated in splitless mode, and the sample injection volume was set as 1 μL for all the samples. Other instrumental parameters, including the temperatures of inlet and detector, as well as the oven temperature program, varied according to the respective derivative methods, and the details are described below in Section 2.3 and Section 2.5.

### 2.3. Screening of Derivative Methods

#### 2.3.1. Esterification

It has been reported that PFCAs can be esterified with small molecular alcohols under acid catalysis to produce volatile esters, which can be detected by GC [24]. In this study, we investigated the esterification efficacies of PFCAs using various alcohols as derivatization reagents. Twenty microliters of the PFCAs stock solution (50 mg L^−1^) was added into 1 mL of methanol, ethanol, and *iso*-propanol, respectively. Then, concentrated H_2_SO_4_ was added to a final concentration of 5% (*v*/*v*). The mixture was shaken at room temperature overnight on an orbital shaker. The PFCA esters were extracted twice by adding 2 mL of *n*-hexane and 0.5 mL of water. The organic phases were combined and further extracted with 0.5 mL of water to remove the acid residual. Then, approximately 20 mg of Na_2_SO_4_ was mixed with the organic extract to remove traces of water. Finally, the extract was concentrated to 1 mL under gentle nitrogen gas flow before GC analysis. The GC injector temperature was set at 220 °C. The oven temperature program was set as follows: the initial temperature of 40 °C was kept for 8 min, then it rose to 130 °C at 5 °C min^−1^, and finally, to 220 °C at 30 °C min^−1^, kept for 5 min.

#### 2.3.2. Amidation 

The method employed is according to the description of the previous report with modifications [26]. Twenty microliters of PFCAs standards (50 mg L^−1^) were diluted with 9 mL distilled water, and the pH was adjusted to 1.0 with 1 N HCl before the addition of 1 g NaCl and 7 mL of ethyl acetate. After the addition of 0.1 mL of 2,4-difluoroaniline and 0.1 mL of *N*,*N*-dicyclohexylcarbodiimide solutions (both stock solutions are 0.1 M dissolved in DCM), the mixture was shaken for 1 h at 200 rpm under ambient temperature. Another one gram of NaCl was dissolved into the mixture, and the ethyl acetate phase was collected. The aqueous phase was re-extracted with an additional 3 mL of ethyl acetate. The combined ethyl acetate extracts were sequentially washed with 1 mL of 1 N HCl, saturated NaHCO_3_ and NaCl solutions. The final extract was dehydrated with anhydrous Na_2_SO_4_ and evaporated under nitrogen gas flow. The sample was finally set to a volume of 0.5 mL in *n*-hexane and analyzed using GC-μECD. In the derivatization and extraction procedures, different organic solvents, including *n*-hexane, DCM, diethyl ether, and ethyl acetate, were used as co-solvent, and ethyl acetate turned out to show the highest derivatization efficiency. Hence, ethyl acetate was used as a co-solvent for derivatization thereafter. The GC injector temperature was set at 300 °C. The oven temperature was programmed as follows: the initial temperature was set at 40 °C and kept for 2 min before it rose to 90 °C at 4 °C min^−1^, kept for 3 min, and finally, to 300 °C at 50 °C min^−1^, kept for 5 min.

### 2.4. Optimization of the Derivative Procedures

The amidation procedure in Section 2.3.2 was further optimized by using the orthogonal design experiments approach. The factors and levels are shown in Table 2, where 7 factors and 3 levels of each factor were investigated. The factors that might affect the amidation efficiency include the pH (A), the mass of sodium chloride (B), the initial volume of ethyl acetate (C), the dose of DFA (D), the dose of DCC (E), temperature (F), and reaction time (G). Orthogonal series of derivative experiments were designed by using the Statistical Product and Service Solutions (SPSS) software, and the generated L_18_ (3^7^) orthogonal table was shown in Appendix A. Eighteen experiments in total were conducted according to Appendix A, followed by the extraction, wash, and concentration procedures, as described in Section 2.3.2. The peak area of the PFCAs derivatives generated by the GC detector was adopted as the evaluation index. 

### 2.5. Optimization of Instrumental Conditions

During all the optimization processes, the nitrogen gas flow rate was constant, and the inlet was fixed at splitless mode. Other instrumental parameters, including the temperatures of inlet and detector, as well as the oven temperature program, were systematically optimized. First, the inlet temperatures were tested at 200, 230, 260, 280, and 300 °C, with all the other instrumental conditions fixed. Then, the temperature of the μECD detector was set at 200, 230, 260, 280, 300, and 320 °C, respectively, with the optimal inlet temperature. Finally, with temperatures of the inlet and detector fixed at optimal values, the program of the oven temperature was further investigated. Three programs were tested, namely TP1, TP2, and TP3, with the details listed as following: 

**TP1:** The initial temperature of 50 °C was kept for 1 min, then it rose to 150 °C at 10 °C min^−1^, where kept for 2 min, and finally to 300 °C at 30 °C min^−1^, kept for 2 min;

**TP2:** The initial temperature of 50 °C was kept for 2 min, then it rose to 120 °C at 5 °C min^−1^, where kept for 2 min, and finally to 300 °C at 20 °C min^−1^, keep for 5 min;

**TP3:** The initial temperature of 50 °C was kept for 1 min, then it rose to 80 °C at 20 °C min^−1^, where kept for 2 min, then it rose to 150 °C at 10 °C min^−1^, where kept for 0 min, and finally to 300 °C at 50 °C min^−1^, kept for 2 min.

### 2.6. Extraction and Cleanup of PFCAs from Water Samples

The extraction and cleanup of PFCAs in water were performed using a solid phase extraction method with WAX cartridges (60 mg, 3 cc, ANPEL, Shanghai, China). The extraction and cleanup method was modified according to previous reports [27]. WAX SPE cartridges were conditioned successively by 6 mL of 0.1% NH_4_OH methanol solution, 6 mL methanol, and 6 mL distilled water, respectively. Five hundred milliliters of distilled water were spiked with 10 μL 100 mg L^−1^ PFCAs mixed standards and loaded onto the cartridge at a rate of 100–150 mL h^−1^. After sample loading, the cartridges were washed with 6 mL of 2.5 M ammonium acetate buffer (pH = 4) and drained to remove the water in the cartridges. Analytes were eluted sequentially by 10 mL each of 0.1%, 1%, and 10% NH_4_OH methanol solutions, and the elutes were collected separately and marked as 1-1, 1-2, and 1-3, respectively. These elutes were reduced to 2 mL under a stream of high purity nitrogen. After mixing with 2 mL of distilled water, the volume of the extract was further reduced to below 2 mL to remove the methanol residual as possible. Then the volume was adjusted to 5 mL using distilled water, and derivatization was performed following the optimal procedures developed in Section 2.4. Furthermore, 1% of NH_4_OH methanol solution was tested for cartridge conditioning, with all the other steps identical with the method described above, and the eluates by 0.1%, 1%, and 10% NH_4_OH were marked as 2-1, 2-2, and 2-3, respectively.

### 2.7. Quality Assurance and Quality Control

Reagent controls using blank solvent were performed following exactly identical procedures with the spiked standard samples during all the experiments, including the screening of derivatization methods, the optimization of the amidation method, as well as the extraction procedures, in order to confirm the interferences of various solvent and reagents. Moreover, all the experiments were conducted in triplicates. For the quantification of PFCAs, ten microliters of 5 mg L^−1^ pentachloronitrobenzene (PCNB) was added to the sample as internal standard just prior to GC injection in order to normalize the instrumental responses. Calibration curves of PFCAs were established by spiking the water with increasing concentrations of standards, followed by derivatization procedures. Limit of detection (LOD) was developed based on the following method: First, the signal/noise ratios (S/N) of the analytes were calculated, and the LODs (concentrations corresponding to S/N = 3) were estimated. Then a specific calibration curve of PFCAs was established following the deviation and instrumental procedures by analyzing spiked samples with PFCAs concentrations in the range of the estimated LODs. The standard deviations (σ) of the residual or y-intercept and slope of the regression line were calculated based on the established calibration curves, and the LODs of each PFCAs congeners were developed according to Equation (1) [28]. Regarding the extraction of PFCAs in water samples, solvent blanks, and spiked experiments were carried out in triplicates, and the recoveries of the target analytes were evaluated. The impact of the matrix effect on the recoveries of target PFCAs was also tested by using surface water samples collected from a local lake, and the composition of the surface water sample was listed in Appendix A.
LODs = 3.3σ/slope(1)


## 3. Results and Discussion

### 3.1. Screening of Appropriate Derivative Method

Presently, there are two derivatization methods available for the detection of PFCAs using GC, namely the esterification method using alcohol under acid-catalysis, and the anilides method. According to the literature [24], the mechanism of the esterification was described by Equation (2). The carboxyl group of the PFCAs can react with various alcohols to produce corresponding esters under the catalysis of acid. In this work, three alcohols were tested for their efficacies of derivatizing PFCAs. As shown in Figure 1a, sharp peaks, which were attributed to the PFCAs esters, appeared in the GC-μECD chromatograms after the esterification processes using the alcohols, and the peaks were identified by single PFCA standards following the same derivatization protocols. However, none of the three alcohols could achieve derivatization of all the nine congeners of PFCAs. In the case of methanol, the methyl esters of PFHpA, PFOA, PFNA, PFDA, PFUnA, and PFDoA were detected, with PFBA, PFPeA, and PFHxA lost. Ethanol could esterify PFHxA, PFHpA, PFOA, PFNA, PFDA, PFUnA, and PFDoA, except PFBA and PFPeA. *Iso*-propanol could achieve the esterification of most PFCAs, with PFBA as an exception. Nevertheless, the responses of the peaks by *iso*-propanol were much lower compared with those generated by methanol and ethanol, probably due to the relatively lower esterification efficiency caused by the steric hindrance in the molecular structure of *iso*-propanol [29,30]. Therefore, it can be concluded that the PFCAs esters approach was only feasible for the derivatization of PFCAs with a longer carbon chain (>C6), but inadequate for short-chained PFCAs, such as PFBA and PFPeA. Similar observations are also reported in a previous study [31]. Although long-chained PFCAs, such as PFOA, are generally more concerned because of their bio-accumulative potential, the short-chained PFCAs cannot be ignored, especially when there are reports that PFBA was detected with extremely high concentrations in the environment as a result of its extensive production and application as the substitute of PFOA or PFOS [32]. Therefore, based on the above analysis, the esterification method was inadequate for the simultaneous determination of the nine target PFCAs.
(2)$${\mathrm{CF}}_{3}{\left({\mathrm{CF}}_{2}\right)}_{n}\mathrm{COOH}+\mathrm{ROH}\;\overset{H_{2}SO_{4}}{\to }\;{\mathrm{CF}}_{3}{\left({\mathrm{CF}}_{2}\right)}_{n}\mathrm{COOR}+H_{2}O$$


The PFCAs anilides method was, thus, tested subsequently. The carboxyl groups of PFCAs can be converted into PFCAs anilides in the presence of 2,4-difluoroaniline (DFA) as a reactant and *N*,*N*′-dicyclohexylcarbodiimide (DCC) as a dehydrating agent (Equation (3)) [33]. As shown by Figure 1b, the height of the peaks generated by the PFCA anilides were approximately two times those of PFCA esters, which suggests the anilides method exhibits higher sensitivity for the detection PFCAs, when compared with the esterification method. This might be attributed to the higher derivatization efficiencies of the amidation process compared with the esterification approaches because the p–π conjunction structure of the amide bond makes it more stable than the corresponding ester bond, and the hydrolysis of amide bond is generally much more difficult than esters [34]. Another possible reason for the enhanced responses is that the produced PFCAs 2,4-difluoroanilides introduce more electron donating atoms, such as fluoride and nitrogen, into the analyte molecules, which is beneficial for improving the responding factors of the analytes in the μECD detector. More importantly, all nine target PFCAs were successfully detected by using this amidation method, which is of great advantage over the esterification method. Therefore, the PFCAs anilides approach was adopted as the final derivatization method in this study, and all the derivatization processes mentioned in this article hereafter refer to this method, unless otherwise stated.
(3)$${\mathrm{CF}}_{3}{\left({\mathrm{CF}}_{2}\right)}_{n}\mathrm{COOH}+C_{6}H_{3}F_{2}{\mathrm{NH}}_{2}\;(\mathrm{DFA})\;\overset{\mathrm{DCC}}{\to }\;{\mathrm{CF}}_{3}{\left({\mathrm{CF}}_{2}\right)}_{n}{\mathrm{CONHC}}_{6}H_{3}F_{2}+H_{2}O$$


### 3.2. Optimization of the Amidation Procedure

The derivatization procedures using the amidation method were further optimized in order to obtain better performance. First of all, the effect of different organic solvents, such as co-solvent with water for derivatization, was investigated. As shown in Appendix A, the use of *n*-hexane as a co-solvent gave no obvious PFCAs peaks, and diethyl ether as a co-solvent for amidation produced relatively higher PFCAs responses than *n*-hexane. When derivatization reactions were conducted in DCM or ethyl acetate, the peak responses of the PFCA derivatives were greatly improved compared with the case of diethyl ether. Nevertheless, the responses of PFCAs derivatives in the DCM system decreased with the increasing length of the carbon chain of PFCAs, and the derivatization efficiencies were insufficient for PFCAs with more than eight carbon atoms. In contrast, ethyl acetate showed relatively higher derivatization efficiency for all the target compounds. Thus, ethyl acetate was chosen as the final co-solvent for the PFCAs anilides derivatization system.

Seven other factors possibly affecting the efficiency of amidation reactions, including pH, mass of sodium chloride, initial volume of ethyl acetate, dose of DFA, dose of DCC, temperature, and reaction time, were investigated with orthogonal design experiments. According to the design listed in Appendix A, eighteen experiments were carried out, and the peak areas of PFBA in the GC chromatograms were adopted as the index for evaluation because other PFCAs showed similar trends with PFBA. Extremum differences (R) analysis was carried out in optimization of the factors that affect the PFCAs anilides production, with the impact of factor levels on the peak area of the anilide products shown in Figure 2a, and the R values of different factors shown in Figure 2b. According to the distribution of R, among all the seven factors investigated, F (temperature), D (DFA dose), and A (pH) impacted the derivatization reaction the most, indicated by their bigger R values than the others. They were followed by C (volume of ethyl acetate used as co-solvent for amidation) and G (the reaction duration). In contrast, factor B (the mass of NaCl in the initial reaction mixture) and E (the dose of DCC as the catalyst for the PFCA anilides formation) had less impact on the derivatization efficiencies (Figure 2b). The optimal levels of the factors could be acquired by checking the peak area of the PFCA derivatives at different levels. The outcome followed the order of A_1_ > A_2_ > A_3_, B_2_ > B_3_ > B_1_, C_2_ > C_1_ > C_3_, D_3_ > D_2_ > D_1_, E_2_ > E_3_ > E_1_, F_1_ > F_2_ > F_3_ and G_1_ > G_3_ > G_2_. Therefore, the optimum derivatization condition is the combination of the optimal level of all the factors: A_1_B_2_C_2_D_3_E_2_F_1_G_1_, i.e., five milliliters of PFCAs water solution was acidified to pH 1 with 1 N HCl, then 0.2 g of sodium chloride, 5 mL of ethyl acetate, 0.4 mL of 2,4-difluoroaniline, and 0.2 mL of *N*,*N*-dicyclohexylcarbodiimide solutions (both 0.1 M in DCM) were added, and the mixture was vibrated in an orbital shaker at 20 °C and 200 rpm for 0.5 h to complete the amidation reaction.

### 3.3. Optimization of GC Parameters

The instrumental parameters of GC-μECD might have significant influences on the responses and separation of target compounds. Here, we investigated the influence of three parameters on the detection of PFCAs anilides, namely the inlet temperature, the detector temperature, and the program of the oven temperature. The composition of the nine PFCAs is complex, and the boiling points of the components are different. Thus, choosing the appropriate inlet temperature and detector temperature is necessary for the sensitive and accurate detection of all the PFCAs components. The peak height of the PFNA derivative was chosen as the index for comparison because PFNA is a medium-chain perfluorinated compound and representative among all the PFCA congeners. As shown in Appendix A, when the inlet temperature was elevated from 200 to 300 °C, the peak height of the PFNA derivative increased first and fell thereafter, with the maximum peak area that emerged at 260 °C. Therefore, the inlet temperature of 260 °C was adopted in the following experiments. With respect to the μECD detector temperature, the peak height of the PFNA derivative increased constantly along with the increase of temperature from 230 to 300 °C, whereas the peak height stopped to enhance at temperatures above 300 °C (Appendix A). In order to maximize the peak height or the sensitivity of the instrument and minimize the energy consumption, as well as the adverse impacts on the detector module and GC column caused by the extremely high temperature, the final temperature of the detector was set at 300 °C. 

In addition, the oven temperature program would pose impacts on the shape and separation of analyte peaks. As can be seen from Figure 1b, the oven program already achieved good peak shapes and separation degrees of the PFCAs components; however, the run time for a single analysis was 22 min, and we hope to shorten the analysis time without deterioration of the shapes and separation of the peaks. Three oven programs were tested (TP1, TP2, and TP3 with their details described in Section 2.5), and the resulting GC chromatograms were shown by Appendix A. It can be found that all the three heating procedures could achieve good shapes and separation of all the nine PFCA anilides. The retention time of PFBA, the first detected PFCA congener in the chromatogram, followed the descending order of TP2 (10.33 min), TP1 (10.22 min), and TP3 (5.77 min). And the retention time of PFDoA, which was the last detected component, followed the decreasing order of TP2 (20.52 min), TP1 (20.41 min), and TP3 (10.42 min). Additionally, the response value generated by the TP3 program was apparently higher compared with the other two programs; for instance, the response value of PFHxA, the highest response value among the nine PFCAs analogs, was 165,153.08 Hz for TP3, higher than those for TP1 (80,175.47 Hz) and TP2 (94,307.92 Hz). This result indicated that the TP3 could shorten the analysis time effectively without deteriorating the peaks’ shape and their complete separation and could simultaneously improve the peak responses of the PFCA anilides to achieve a higher sensitivity of detection. Therefore, TP3 was selected as the final oven temperature program for PFCA anilides analysis.

### 3.4. Validation of the Derivatization Method

The amidation method was first validated by establishing the calibration curves of the PFCAs. According to the optimized derivatization procedure and instrumental conditions, pure solvent spiked with different amounts of PFCAs standards were transformed into PFCAs anilides, and the theoretical final concentrations of each PFCA analog were designed to be 0.1, 0.2, 0.4, 0.6, 0.8, 1, and 2 mg L^−1^ in the 1 mL *n*-hexane solutions. Calibration curves were established by plotting the peak areas of each PFCA congener (y) against their theoretical concentrations (x), and the fitted linear equations are listed in Table 3. All the linear curves fitted well (R^2^ > 0.99) and could meet the requirement for quantification. However, for theoretical final concentration above 2 mg L^−1^, poor linearity was obtained for all the nine PFCAs (data not shown), indicating that the upper limit of the linear range for this amidation derivatization and GC-μECD method should be approximately 2 mg L^−1^.

Then the LODs of this method were evaluated. As described in Section 2.7, specific calibration curves were tested using PFCA standards with theoretical final concentrations of 0.8, 2, 4, 8, 16, 24, 32, and 40 μg L^−1^. Then, linear regression analyses of the plots on the calibration curves were conducted to obtain the necessary parameters for LODs calculation, including the standard deviations (SD) of the residual and y-intercept, the slope of the fitted linear regression. The parameters obtained and the LODs calculated based on the corresponding parameters were listed in Appendix A. The LODs of PFCAs ranged from 2.12 to 11.76 μg L^−1^ when the SD of residuals was used for calculation, and they fell in the range of 1.14 to 6.32 μg L^−1^ when the SD of the intercepts was used. The LODs of this method in the present study were slightly higher than the instrumental detection limits of PFAAs using the HPLC-MS-MS method, which approximately ranged from 0.01 to 10 μg L^−1^, calculated according to the limit of detection reported and the corresponding sample extraction procedures [35,36]. Nevertheless, the problem of higher LODs can be tackled by employing proper extraction and pre-concentration of the PFCAs in water samples before derivatization and GC analysis. For instance, when PFCAs in 1L of water was extracted and concentrated to the final volume of 100 μL for GC analysis, the detection limits would be ca. 0.1–0.6 ng/L, which are significantly lower than the reported PFCAs concentrations (several to several hundred ng L^−1^) in natural waters [32,37,38,39]. The extraction and pretreatment procedures of water samples before derivatization will be further studied in Section 3.5 of this article. More importantly, the GC method is much more cost effective compared with the HPLC-MS/MS method, based on the expense evaluation of the two different methods. On average, it would save ca. $28.7 to $66.2 per sample by using the GC method (Appendix A), and the extra time for the derivatization process is only approximately 1 h. Therefore, we can strike a tradeoff between the money and time cost for PFCAs analysis by employing the GC method.

### 3.5. Pretreatment Procedures of Water Sample and Validation of the Whole Method

As stated above, the water samples have to be pretreated in order to extract and concentrate the target PFCAs. Solid phase extraction (SPE) was widely used for the extraction and cleanup of trace pollutants in water samples because of its advantages, such as high enrichment factors, easy handling, less organic solvent consumption, and secondary pollution [36,40,41]. According to the literature, two types of SPE cartridges, namely HLB (hydrophilic-lipophilic balanced sorbents) and WAX (mix-mode weak anion exchanger), are frequently used for the extraction of PFCAs from surface water. Comparatively, WAX cartridges show higher extraction efficiency and lower matrix effect, especially for the extraction of PFCAs with different chain lengths [39,42,43]. Therefore, in this study, WAX cartridges were adopted, and the extraction procedures were optimized. In terms of elute NH_4_OH concentrations, the chromatograms showed that all the PFCAs were detected in the fraction eluted by 10 mL of 0.1% NH_4_OH methanol solution (1-1 and 2-1 in Appendix A). Meanwhile, no target compound was detected in the subsequent fractions eluted by higher concentrations of NH_4_OH (10 mL of 1% and 10% NH_4_OH, shown by chromatograms of 1-2, 1-3, 2-2, and 2-3 in Appendix A). Therefore, the 0.1% NH_4_OH methanol solution was sufficient to elute the target compounds. With respect to the NH_4_OH solution for cartridge conditioning before sample loading, we found from Appendix A that the WAX cartridge pre-conditioned using 1% NH_4_OH showed lower recoveries for most of the PFCA analogs when compared with those acquired by using 0.1% NH_4_OH solution. In addition, the use of 1% NH_4_OH for cartridge conditioning resulted in the abnormal rise of the peak corresponding to PFDoA, which might be due to the interference caused by the high concentration of NH_4_OH. Therefore, it is obvious that the 0.1% NH_4_OH solution is sufficient to activate the anion exchange active sites of the sorbent in the WAX cartridges. The final SPE conditions were determined as follows: 0.1% NH_4_OH methanol solution for WAX cartridge pre-condition in advance of sample loading, and the identical 0.1% NH_4_OH solution for PFCAs elution after sample loading.

In order to assess the recoveries of the PFCAs after the SPE procedure, a standard sample at identical theoretical concentrations of PFCAs with 1-1 was directly derived into anilides without SPE treatment (standard, Appendix A), and it is noteworthy that the peak heights of most PFCA anilides in 1-1 were only half of those in the “standard”. There might be three possible reasons: (1) the SPE cartridge failed to retain all the target compounds when water samples were loaded; (2) the PFCA targets were not completely eluted from the sorbent; (3) the PFCAs evaporated during the concentration process by nitrogen gas. Reason (2) can be excluded because no PFCA was found in the subsequent elute using methanol with elevated ammonium hydroxide concentrations. Cause (1) was neither likely because the sorbent volume was excessive for the loaded amount of PFCAs. Therefore, it was most likely that PFCAs evaporated under nitrogen gas flow. In order to elucidate our hypothesis, PFCAs standard spiked in 10mL of 0.1% NH_4_OH methanol solution was concentrated by nitrogen flow, and subsequently derivatized and analyzed by GC. As expected, the sample treated by nitrogen gas (STD-N_2_) generated much lower peak responses of PFCA anilides when compared with the counterpart without treatment by nitrogen gas flow (STD, Appendix A). This is supported by the saturated vapor pressures of PFCAs. For instance, at 25 °C, the vapor pressures of PFOA and PFBA were predicted to be about 70 and 849 Pa, respectively, by using the US Environmental Protection Agency’s EPI Suite™. Whereas some typical POPs possess much lower vapor pressures, for instance, 5.12 × 10^−4^ Pa for α-HCH, 5.12 × 10^−4^ Pa for *p*,*p*′-DDT, 1.3 × 10^−4^ Pa for PCB-180, and 4.68 × 10^−7^ Pa for BDE-183. The saturated vapor pressures of PFCAs are more than five orders of magnitude higher than those of other semi-volatile organic pollutants, so it is difficult to avoid loss during the concentration process of the elutes by nitrogen gas flow. 

However, the problem of PFCAs evaporation during sample pretreatment would result in low recoveries of the targets, thus, hindering the accurate quantification of PFCAs in water samples. In order to tackle this problem, we prepared 500 mL of water samples spiked with a gradient amount of PFCAs standards, and these water samples were extracted, concentrated, derivatized, and quantified by GC. Then, calibration curves were established for all the PFCA congeners based on these standard samples (Table 4). Quantification using these calibration curves could normalize the low recoveries of the PFCAs caused by the volatilization of target compounds during the pretreatment process. It can be seen that the linearity of the standard curves can meet the quantitative demand, with R-squared values greater than 0.99 for all the PFCA analogs. Based on these standard curves, we managed to validate the whole method for PFCAs determination in water samples by conducting experiments, including the method blanks (samples prepared using pure water), spiked blanks (standards spiked into pure water), matrix blanks (surface water samples collected from local lake), and spiked matrices (standards spiked into the surface water samples). No PFCA was detected in the method blank samples (data not shown). The recoveries of PFCAs in the spiked blanks ranged from 62% to 118%, with the relative standard deviation (RSD) of triplicates in the range of 8–19%. At the same time, the recoveries of the PFCAs in the spiked matrices fell in the range of 57–117%, with RSD values ranging from 6–17% (Table 5). The recoveries of the target compounds are generally acceptable and meet the requirement for quantification of trace pollutants in water. This suggests the good accuracy (reflected by the recoveries of PFCAs in spiked samples) and precision (characterized by the RSD of triplicates less than 20%) of the method [44].

The method developed in our present study was further compared with other methods. The currently available methods for the detection of PFCAs mainly included GC and HPLC-MS-MS based ones. The GC methods always required derivatization, and the generation of esters was the most frequently employed. Unfortunately, as compared in our present study, the esterification method was only valid for the detection of C7–C12 PFCAs and unable to derivatize shorter chained PFCAs. On the contrary, the 2,4-difluoroanilide method could analyze C4–C12 PFCAs, with higher sensitivity than the esterification method. The HPLC-MS-MS methods were most popular nowadays for the determination of PFCAs with high sensitivity and selectivity, but the most concerning issue of these methods lies in the high expense of the instrument, which was always poorly accessible and unaffordable for routine analysis, especially in developing parts of the world. From this point of view, our method is much cheaper, and readily applicable thanks to the common availability and low cost of the GC instruments. Moreover, the HPLC-MS-MS methods had other drawbacks, such as the matrix suppression effects caused by co-extractive materials, and the false positive results caused by the background signals due to the leak of PFCAs from the fluorinated polymeric parts of the instrument. These problems could also be avoided by employing our present GC method. Moreover, when compared with the method reported by Scott and coworkers, which also involved the production of 2,4-difluoroanilide derivatives [26], the SPE method for water pretreatment in our study is apparently more suitable for high throughput analysis of batch samples, whereas the rotary evaporation method or the XAD-7 resin method they employed can only treat a single sample at once and, thus, would be too time- and labor-consuming for batch samples. In addition, we used only 10 percent of the reagents dose in those by Scott et al. and avoided the toxic toluene in the PFCAs derivatizing procedure to achieve a similar detection limit; thus, our method is more ecofriendly. More importantly, Scott’s method only showed good recoveries of the C7–C9 PFCAs, but our method obtained good recoveries of all the C4–C12 PFCAs (Appendix A).

## 4. Conclusions

Here, we demonstrated that the amidation method was more effective for the derivatization of PFCAs (C4–C12) compared with esterification. We managed to systematically optimize the amidation procedures, the instrumental parameters of GC, and integrated the derivatization procedure with the SPE procedure to develop an intact method for the quantification of trace PFCAs in surface water samples. This method is valid for application, with good accuracy and precision. Therefore, we believe the GC-based detection method developed in the present study could serve as a promising alternative of the HPLC-MS/MS method and provide a cost-effective solution for the determination of PFCAs in various environmental matrices.

## Figures and Tables

**Figure 1 ijerph-17-00100-f001:**
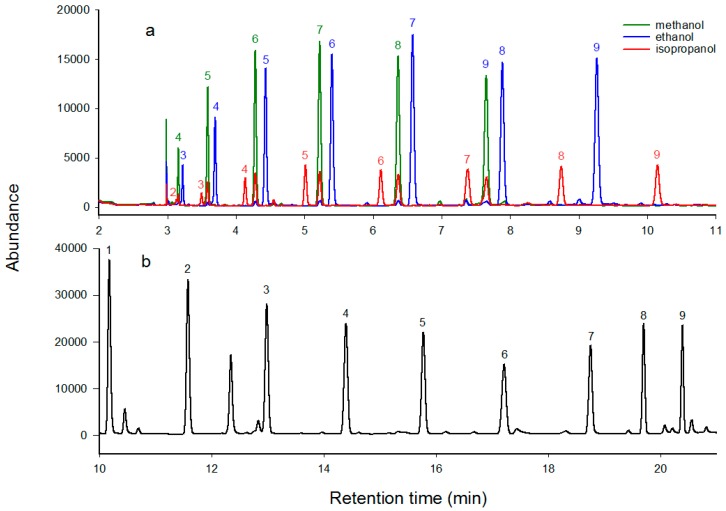
Gas chromatography-micro-electron capture detector (GC-μECD) chromatograms of (**a**) PFCAs esters produced by using three different alcohols; (**b**) PFCAs anilides derivatized by using 2,4-DFA and DCC. Peak identification: (**1**) PFBA, (**2**) PFPeA, (**3**) PFHxA, (**4**) PFHpA, (**5**) PFOA, (**6**) PFNA, (**7**) PFDA, (**8**) PFUnA and (**9**) PFDoA.

**Figure 2 ijerph-17-00100-f002:**
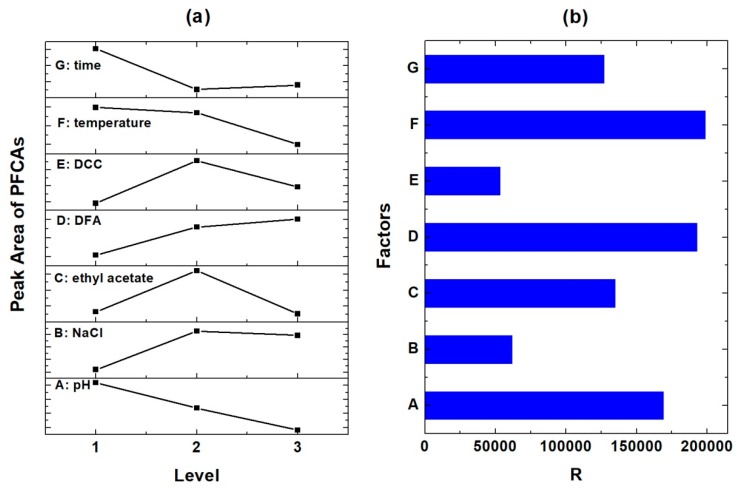
Results of the orthogonal design experiments: (**a**) the trends of PFCAs anilides peak area along with the varied levels of the factors; (**b**) comparison of the extremum differences (R) between the factors.

**Table 1 ijerph-17-00100-t001:** Detailed information of the target analytes.

Compound	Formula	Acronym	CAS No.
Perfluorobutyric acid	CF_3_(CF_2_)_2_COOH	PFBA(C4)	375-22-4
Perfluoropentanoic acid	CF_3_(CF_2_)_3_COOH	PFPeA(C5)	2706-90-3
Perfluorohexanoate acid	CF_3_(CF_2_)_4_COOH	PFHxA(C6)	307-24-4
Perfluoroheptanoate acid	CF_3_(CF_2_)_5_COOH	PFHpA(C7)	375-85-9
Perfluorooctanoate acid	CF_3_(CF_2_)_6_COOH	PFOA(C8)	335-67-1
Perfluoronanoate acid	CF_3_(CF_2_)_7_COOH	PFNA(C9)	375-95-1
Perfluorodecanoate acid	CF_3_(CF_2_)_8_COOH	PFDA(C10)	335-76-2
Perfluoroundecanoate acid	CF_3_(CF_2_)_9_COOH	PFUnA(C11)	2058-94-8
Perfluorododecanoate acid	CF_3_(CF_2_)_10_COOH	PFDoA(C12)	307-55-1

**Table 2 ijerph-17-00100-t002:** Factors-levels of the orthogonal experiments design for perfluoroalkyl carboxylic acids (PFCAs) anilides derivatization.

Factor	Levels		
	1	2	3
pH (A)	1	3	7
NaCl g^−^^1^ (B)	0	0.2	0.5
ethyl acetate mL^−^^1^ (C)	2	5	7
DFA: ethyl acetate (*v*/*v*, D) ^a^	0.02	0.04	0.08
DCC: ethyl acetate (*v*/*v*, E) ^a^	0.02	0.04	0.08
temperature °C^−1^ (F)	20	40	60
time h^−1^ (G)	0.5	1	2

^a^ The concentrations of 2,4-difluoroaniline (DFA) and *N*′-dicyclohexylcarbodiimide (DCC) stock solutions are 0.1 M dissolved in dichloromethane (DCM).

**Table 3 ijerph-17-00100-t003:** Equations and R-squared values of the linear regression for 9 PFCAs, where y represents the peak areas of PFCA anilides, and x stands for the theoretical concentrations of corresponding PFCAs in mg L^−^^1^. The theoretical concentrations are 0.1, 0.2, 0.4, 0.6, 0.8, 1, and 2 mg L^−1^.

Compounds	Equation	R^2^
PFBA	y = 278018x + 9839	0.9988
PFPeA	y = 240734x + 1186	0.9984
PFHxA	y = 233148x + 10534	0.9911
PFHpA	y = 156992x + 2147.3	0.9959
PFOA	y = 87682x + 25658	0.997
PFNA	y = 29998x + 4876.6	0.9924
PFDeA	y = 92244x + 27935	0.9936
PFUnA	y = 75254x + 10022	0.9933
PFDoA	y = 74725x + 9578.2	0.9934

**Table 4 ijerph-17-00100-t004:** Regression equations and the R squared values of 9 PFCAs anilides in spiked water samples, which were pretreated by SPE and concentrated by nitrogen gas flow before amidation. Here, y represents the peak area ratio of PFCAs anilides and the internal standard PCNB, and x stands for the theoretical concentrations of PFCAs in the final 1 ml of *n*-hexane.

Compounds	Equation	R^2^
PFBA	y = 2.8575x − 0.766	0.9901
PFPeA	y = 2.5306x − 0.5246	0.9913
PFHxA	y = 2.391x − 0.438	0.9942
PFHpA	y = 2.1184x − 0.5477	0.997
PFOA	y = 1.1813x − 0.1267	0.9902
PFNA	y = 0.5568x − 0.0075	0.9944
PFDeA	y = 0.3992x − 0.0416	0.996
PFDoA	y = 0.2853x − 0.0392	0.9917
PFUnA	y = 0.2566x + 0.1367	0.9913

**Table 5 ijerph-17-00100-t005:** Validation of the SPE/GC-μECD method for PFCAs quantification in water samples. Indices include the recoveries and relative standard derivatizations (RSD) of the PFCA congeners in spiked blanks and matrices.

Compound	Spiked Blanks		Spiked Matrices	
	Recovery	RSD	Recovery	RSD
PFBA	62%	11%	57%	11%
PFPeA	79%	15%	65%	17%
PFHxA	76%	17%	70%	11%
PFHpA	75%	18%	76%	14%
PFOA	92%	15%	70%	10%
PFNA	113%	11%	117%	7%
PFDeA	88%	18%	111%	15%
PFDoA	83%	19%	109%	6%
PFUnA	118%	8%	104%	16%

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
