# Peer review of "Cost-Effective Detection of Perfluoroalkyl Carboxylic Acids with Gas Chromatography: Optimization of Derivatization Approaches and Method Validation"

_ijerph, 2019, doi:10.3390/ijerph17010100_

Round 1

Reviewer 1 Report

The main objective of this paper is the quantification of perfluoroalkyl carboxylic acids with gas chromatography. The contribution of this study is clear, however, more emphasis has to be done on the advantages over other methods such as those based on HPLC. The authors should include more information on the composition of the surface water samples collected from the local lake.

Reviewer 2 Report

Li and Sun have described a method for detection of perfluoroalkyl carboxylic acids by using GC-MS. The main focus in the modification of a previously reported derivatization method reported by Scott and coworkers (Environ. Sci. Technol. 2006, 40, 20, 6405-64100). The work is interesting and can be published in IJERPH contingent on a minor to major revision. I would be useful if the authors compare their method to the original method developed by Scott et al. which involves preparing the 2,4-difluoroanilide derivatives of the acids. Adding a table might be useful.  On page 7 (equation 3), I suggest that the authors write the name of C13H22N2 as there may be isomers of this formula.
